# Analysis of Acute Phase Response Using Acute Phase Proteins Following Simultaneous Vaccination of Lumpy Skin Disease and Foot-and-Mouth Disease

**DOI:** 10.3390/vaccines12050556

**Published:** 2024-05-19

**Authors:** Jiyeon Kim, Danil Kim, Hyoeun Noh, Leegon Hong, Eunwoo Chun, Eunkyung Kim, Younghye Ro, Woojae Choi

**Affiliations:** 1Department of Farm Animal Medicine, College of Veterinary, Seoul National University, Seoul 08826, Republic of Korea; melilsa8037@snu.ac.kr (J.K.); danilkim@snu.ac.kr (D.K.); shgydms97@snu.ac.kr (H.N.); leegon1213@snu.ac.kr (L.H.); 2Farm Animal Clinical Training and Research Center, Institutes of Green-Bio Science and Technology, Seoul National University, Pyeongchang 25354, Republic of Korea; alisonchun@snu.ac.kr (E.C.); ek2426@snu.ac.kr (E.K.); 3Department of Large Animal Medicine, College of Veterinary Medicine, Kangwon National University, Chuncheon 24341, Republic of Korea

**Keywords:** acute phase protein, adverse reactions, foot-and-mouth disease, lumpy skin disease, vaccination

## Abstract

Since 2011, South Korea has implemented biannual vaccinations against foot-and-mouth disease (FMD) and recently, lumpy skin disease (LSD), to mitigate the spread of transboundary animal diseases. However, due to past adverse reactions, potentially linked to acute phase responses from FMD vaccinations, there is hesitancy among Korean livestock farmers regarding new strategies for simultaneous vaccinations against both FMD and LSD. This study was conducted to assess possible adverse reactions to the LSD vaccination by analyzing acute phase proteins (APPs) in three groups: cows vaccinated against FMD (G1-FMDV), LSD (G2-LSDV), and both (G3-FMDV/LSDV). In G1-FMDV, APP levels peaked on day 3 post-vaccination (*p* < 0.001) and returned to baseline. In G2-LSDV, APP levels increased gradually, peaking on day 10 post-vaccination. In G3-FMDV/LSDV, APP levels peaked on day 3 post-vaccination and remained high until day 10 (*p* < 0.001). These results indicate that LSD vaccines trigger a later immune response compared to FMD vaccines, possibly due to different adjuvants. Therefore, a longer follow-up period for monitoring adverse reactions to LSD vaccinations may be required to understand and mitigate potential risks.

## 1. Introduction

Foot-and-mouth disease (FMD) is a highly contagious viral disease caused by the *foot-and-mouth disease virus*, a member of the *picornaviridae* family, which causes vesicles on the oral mucous membrane and interdigital skin of cloven-hoofed animals, including cattle and pigs [1,2,3,4]. The clinical signs of FMD are characterized by excessive salivation and reduced feed intake, and prevention is of vital importance due to the rapid spreading of the disease [1,2,3,4,5]. In South Korea, the first outbreak of FMD occurred in November 2010, rapidly spreading all over the country [6,7]. The emergency FMD vaccination was conducted by the government a month later, and since 2011, the government has adopted a biannual vaccination program (in April and October), which has been similarly conducted in other countries [6,7,8].

Lumpy skin disease (LSD) is a vector-borne viral disease caused by infection with *lumpy skin disease virus*, belonging to the *poxviridae* family [9,10,11,12,13,14,15,16]. LSD originated in Africa in 1929 and is transmitted through vectors, especially arthropods such as ticks, flies, and mosquitoes [9,10,11,12,13,17,18]. The disease has spread from southern Africa to northern Africa over several decades. In 1989, the LSD outbreak in Israel, a Middle Eastern country, spread to Europe, and then to Asia after 2013 [10,11,12,13,17]. LSD can cause nodular skin lesions, lymph node enlargement, mucosal edema of the skin, and productivity loss, including reduced weight gain or milk yield [10,12,13,16,17].

Both FMD and LSD are classified as class 1 livestock infectious diseases, as well as notifiable diseases, in the Republic of Korea, as well as by the World Organization for Animal Health [4,9,19]. Vaccination remains the primary defense, and South Korea has implemented biannual campaigns against FMD [1,6,7,8,10,13,15]. Despite this, vaccine hesitancy exists among farmers because of the fear of adverse effects such as abortion and reduced productivity [5]. The recent emergence of LSD in South Korea in October 2023 has intensified efforts toward vaccination against FMD and LSD, sparking debates regarding the necessity of additional annual vaccinations. According to previous studies, the immune responses to LSD vaccine are mild compared to those noted after the FMD vaccine; however, symptoms including skin lesions, decreased milk yield, fever, and decreased dry matter intake after LSD vaccination have been reported [13,16,17]. In Türkiye, vaccines against FMD and LSD are administered twice per year as part of a control strategy [8]. It has been confirmed that simultaneous administration of both vaccines does not affect antibody production or protective capabilities [8]. However, there has not been a study focused on the immune response to simultaneous vaccination. Despite these assurances, concerns remain among field practitioners regarding possible adverse reactions to the combination of FMD and LSD vaccines.

For the evaluation of the reaction after vaccination, we conducted analyses of acute phase protein (APP), including haptoglobin (HAP) and serum amyloid A (SAA) [20,21,22]. The APPs are increased in acute phase response (APR), which is the defense mechanism of organisms to trauma, surgery, infections, inflammation, or tissue damage [20,21,22]. Specifically, the SAA can increase up to 1000-fold during diseases in the acute phase, and these APPs syntheses and APRs are stimulated by the interleukin-1, interleukin-6, interferon-gamma, etc. [21,22]. In previous studies, the plasma concentrations of SAA and HAP were used for inflammatory disease diagnosis, as non-specific markers of inflammation in dairy cattle [20,21].

The aim of this study is to analyze and compare the characteristics of the acute immune responses, specifically through the evaluation of APP levels, when FMD and LSD vaccines were administered together versus separately.

## 2. Materials and Methods

The study was conducted with 5 Korean Native Cows and 12 Holstein Friesian cows. To determine the effect of each vaccination on the changes in APP level, five cows were administered with the FMD vaccine alone (G1-FMDV), six cows were injected with the LSD vaccine alone (G2-LSDV), and the remaining cows (G3-FMDV/LSDV) were administered both vaccines simultaneously. The FMD vaccine (Himmvac FMD, KBNP Inc., Anyang, Republic of Korea) was injected intramuscularly into the cervical region, 2 mL per head, and the LSD vaccine (Lumpyvac, VETAL Animal Health Products Inc., Adiyaman, Türkiye), 2 mL, was administered subcutaneously into the cervical region. 

Blood samples were collected from the coccygeal vessel with a heparin tube (Lithium Heparin tube, BD Vacutainer, Franklin Lakes, NJ, USA) on days 0, 3, 6, and 10 after vaccination. Plasma was collected within 1 h after blood sampling, under centrifugation at 3000 rpm for 15 min, and stored at −70 °C until analysis. The frozen plasma was thawed at room temperature prior to each test. To confirm changes in plasma APP level, HAP and SAA were analyzed using a commercial ELISA kit (E-10HPT, ICL Inc., Portland, OR, USA; TP 802, Tridelta Development Ltd., Kildare, Ireland). This study was conducted after approval from the Institutional Animal Care Use Committee of Gyeongsangbuk-do Livestock Research Institute (#141).

All results were analyzed using SigmaPlot 15 (Systat Software Inc., Palo Alto, CA, USA). Two-way repeated measures ANOVA, followed by multiple comparisons using a Bonferroni *t*-test, were used to identify significant differences within the group and over time.

## 3. Results

The results of two-way repeated measures ANOVA are presented in Table 1. The results for SAA levels are illustrated in Figure 1. In G1-FMDV, the plasma SAA level was significantly higher on day 3 post-vaccination, compared to day 0 (*p* < 0.001). The plasma SAA levels returned to their initial levels on days 6 and 10 post-vaccination. G2-LSDV showed a tendency for a gradual increase in SAA levels until 10 days after vaccination; however, the difference was not statistically significant (*p* = 0.06). In G3-FMDV/LSDV, SAA levels were significantly elevated at all time-points compared to those at day 0 (*p* < 0.01). Notably, on day 3 post-vaccination, G3-FMDV/LSDV exhibited a significantly higher SAA level than that of G2-LSDV (*p* < 0.001); however, there was no significant difference compared with G1-FMDV (*p* = 0.425). The SAA level on day 10 post-vaccination was significantly higher in G3-FMDV/LSDV3 than in G1-FMDV and G2-LSDV (*p* < 0.05).

The results regarding the HAP levels are presented in Figure 2. A significant elevation in the HAP level was observed only in G3-FMDV/LSDV from day 3 to day 10 post-vaccination (*p* < 0.001). In the comparison between groups, a significantly higher HAP level was confirmed in G3-FMDV/LSDV than in G2-LSDV on day 3 post-vaccination (*p* < 0.01), and the levels tended to be higher than that in G1-FMDV (*p* = 0.064). On day 6 and day 10 post-vaccination, the HAP levels of G3-FMDV/LSDV were significantly higher than those in G1-FMDV and G2-LSDV (*p* < 0.001).

## 4. Discussion

In this study, APP levels increased and reached a high level on day 3 post-vaccination with FMD, as in previous studies, followed by a gradual decline to pre-vaccination levels by day 10 post-vaccination [5]. Conversely, an elevation in APP levels was observed only starting from day 10 post-vaccination of LSD, indicating a delayed immune response with the LSD vaccine. In the group administered both LSD and FMD vaccines simultaneously, the plasma levels of the APPs increased from day 3 post-vaccination due to the FMD vaccine. Also, it exhibited an increase 10 days after vaccination, according to the influence of the LSD vaccine. 

Previous studies have reported various responses following FMD vaccination, including fever, milk yield loss, and gastric disorder, which resolved after approximately 1–3 days [23]. According to Kim et al., white blood cell counts in the blood reached significantly high levels at 3 days after vaccination and decreased gradually [5]. In addition, the plasma APP levels elevated until 3 or 4 days post-vaccination and then returned to initial levels [5]. However, until now, there was no study of blood APP levels after LSD vaccination, but only of clinical signs after vaccination. According to Gari et al., in the LSD challenge experiment in vaccinated cattle, the clinical signs began appearing from 6 to 8 days post-challenge [24]. In addition, the clinical reaction score, encompassing skin lesions, nasal or ocular discharge, fever, loss of appetite, and lymphadenopathy, increased until day 13 and was then maintained for 3 weeks post-challenge [24]. In another study, decreased milk yield was noted between 6–12 days after vaccination, which subsequently recovered to the pre-vaccination milk yield by day 28 post-vaccination [16]. The time when post-vaccination clinical signs appear coincides with the time frame of APP level elevation [5,16,23,24]. In an in vitro study investigating *capripoxvirus* vaccines, by Varshovi et al., the concentration of cytokines, IFN-gamma, and IL-4 increased at 7 days and peaked at 21 days after vaccination [19]. Therefore, based on the timing of the APP level elevation, it can be assumed that the immune response, including mild adverse effects, after LSD vaccination appears later than after FMD vaccination. 

The purpose of vaccination is to acquire immunity, and an appropriate combination of adjuvants can affect the effectiveness of the vaccine [25]. The difference in the elevating timing of acute phase response is assumed because of the difference in the adjuvant used in each vaccine. In the FMD vaccine, which is an inactivated virus vaccine, paraffin oil is added as an adjuvant to increase immunogenicity [3,4,23]. In contrast, LSD vaccine is an attenuated virus vaccine, and a combination of lactalbumin hydrolysate and sucrose is used as a stabilizer, along with an oily emulsion with Montanide as an adjuvant [13,26]. The adjuvant used in the FMD vaccine induces an immune response by increasing monocytic exudation, which causes an inflammatory response similar to that of a pathogen infection [25]. The adjuvant used in the LSD vaccine stabilizes the antigen and maintains antigenic stabilization for a long period [8,12,25]. The adverse effects of LSD vaccine administration have been controversial [11,12,13,14,15,16]. In the FMD vaccine, clinical signs such as decreased dry matter intake or decline in milk yield have been reported 3 days after vaccination [3,23]. The main clinical signs of both vaccinations include swelling or nodules at the injection site [12,13,14,15,25]. This induces the release of pro-inflammatory cytokines, which trigger non-specific activation of the innate immune system and stimulation of adaptive immunity [25]. Although a previous study has shown that simultaneous administration of the sheep pox, goat pox, and FMD vaccine results in sufficient antibody titer, no research has been conducted investigating acute phase responses after both vaccinations [8,19]. This discrepancy in adjuvant action may explain the differential timing of the acute phase responses observed between the two vaccines.

In several studies, fever was observed within 3 days after vaccination for both FMD and LSD [2,6,7,15,19,23]. Fever can be considered as one of the innate immune responses following vaccination. In particular, body temperature above 40 °C has been confirmed for 3 days in FMD vaccination and for 3 to 5 days in LSD vaccination [2,15,19,23]. Mainly, fever is known as the acute phase response and is a systemic defense mechanism against infection [27]. Based on the results of the APP in this study, the timing of APP level elevation did not coincide with the onset of fever. This supports previous research suggesting that fever, which is one of the acute phase responses, and elevated APP levels do not have the same definition [27].

This study confirmed the degree of acute phase response by measuring APP levels after FMD and LSD vaccination over a short period. In South Korea, biannual vaccination has been performed for over 10 years to prevent FMD, and concerns over side effects including abortion, stillbirth, decreased milk yield, and other reproductive disorders due to the vaccine immune response have been raised by farmers [4,5,6,7]. However, because LSD vaccination was conducted for the first time in the Republic of Korea this year, there have been no reports of adverse effects of the vaccine, such as abortion. Therefore, the adverse effects of the vaccine have not been clearly revealed, and its causal relationship remains unclear. Despite the lack of comprehensive data on the adverse effects of the LSD vaccine due to its recent introduction, our findings indicate that post-vaccination clinical signs and APP level elevation are temporally correlated, aiding in the anticipation of vaccine-induced responses [12,13,14,15]. Although the APP level increased 3 days after vaccination in the FMD group, increased 10 days after vaccination in the LSD group. This indicates that the clinical signs after vaccination with LSD may present later than those for FMD vaccination. Therefore, a longer observation period is necessary to adequately assess the adverse effects of the LSD compared to the FMD vaccine.

According to Kim et al., after FMD vaccination, alteration patterns in ruminoreticular temperature and body activity vary, depending on the pregnancy of the experimental animals [1]. However, the present study analyzed APP levels exclusively in non-pregnant cows, which leaves consequent reproductive effects such as stillbirth, delayed ovulation, and abortion in vaccinated pregnant cows unexplored. Moreover, it is necessary to investigate the long-term persistence of the acute phase response caused by simultaneous FMD and LSD vaccinations, as well as their adverse effects. Further research is needed to address these challenges, which will contribute to mitigating potential adverse effects and enhancing vaccine safety and efficacy.

## 5. Conclusions

This study analyzed alterations in APP levels to compare the immune responses elicited by the FMD and LSD vaccinations. An acute phase response was the highest 3 days after FMD vaccination but remained high until 10 days after LSD vaccination. Through this difference in timing of the acute phase response between vaccines, it is estimated that the onset of side effects may be delayed in the LSD vaccination when compared to those in the FMD vaccination. Therefore, it is recommended that the follow-up period after LSD vaccination be extended compared to that employed after FMD vaccination.

## Figures and Tables

**Figure 1 vaccines-12-00556-f001:**
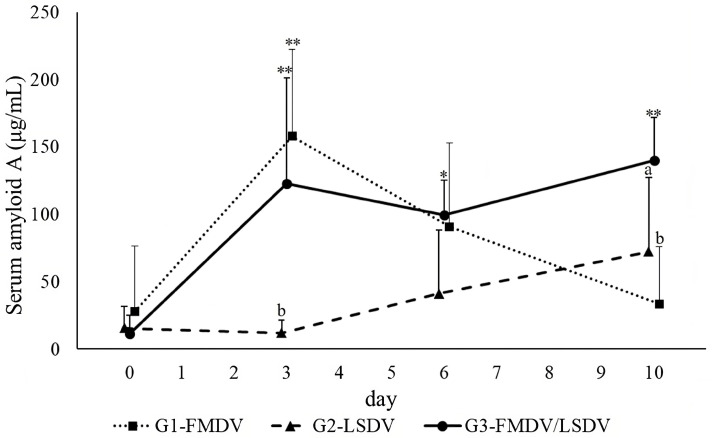
Changes in plasma serum amyloid A (SAA) in plasma in each groups (G1-FMDV, G2-LSDV, and G3-FMDV/LSDV). Data are expressed as means ± standard deviation. A significant difference at each time point is noted for day 0 (*, *p* < 0.05; **, *p* < 0.001), and a significant difference in G1-FMDV and G2-LSDV is observed in comparison with G3-FMDV/LSDV (control) (a, *p* < 0.05; b, *p* < 0.001). G1-FMDV, the group with FMD vaccine administration; G2-LSDV, the group with LSD vaccine administration; G3-FMDV/LSDV, simultaneous vaccination with FMD and LSD.

**Figure 2 vaccines-12-00556-f002:**
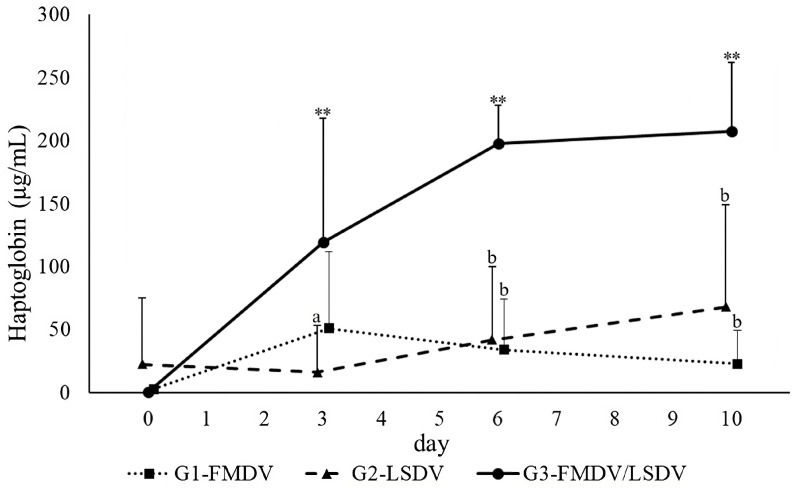
Changes in plasma haptoglobin levels. Data are expressed as means ± standard deviation. A significant difference at each time point is observed in comparison with day 0 (**, *p* < 0.001), and a significant difference in G1-FMDV and G2-LSDV is noted in comparison with G3-FMDV/LSDV (control) (a, *p* < 0.05; b, *p* < 0.001). G1-FMDV, the group with FMD vaccine administration; G2-LSDV, the group with LSD vaccine administration; G3-FMDV/LSDV, simultaneous vaccination with FMD and LSD.

**Table 1 vaccines-12-00556-t001:** Results of the two-way repeated ANOVA with one factor repetition for serum amyloid A and haptoglobin in plasma.

**Serum Amyloid A**					
**Source**	**DF**	**SS**	**MS**	***F*-Ratio**	***p*-Value**
GROUP	2	43,101.063	21,550.531	6.576	0.010
TIME	3	61,193.208	20,397.736	12.107	<0.001
GROUP × TIME	6	67,421.821	11,236.970	6.670	<0.001
**Haptoglobin**					
**Source**	**DF ^1^**	**SS ^2^**	**MS ^3^**	***F*-ratio**	***p*-value**
GROUP	2	151,889.219	75,944.610	19.184	<0.001
TIME	3	85,499.298	28,499.766	13.034	<0.001
GROUP × TIME	6	86,930.508	1448.842	6.626	<0.001

^1^ DF, degree of freedoms; ^2^ SS, sum of squares; ^3^ MS, mean squares.

## Data Availability

Data is contained within the article or Appendix A.

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
