# Peer review of "Analysis of Acute Phase Response Using Acute Phase Proteins Following Simultaneous Vaccination of Lumpy Skin Disease and Foot-and-Mouth Disease"

_vaccines, 2024, doi:10.3390/vaccines12050556_

Round 1
Reviewer 1 Report
Comments and Suggestions for Authors
The authors present an interesting study where they have examined the acute phase protein responses as an indicator of the immune response to an FMDV vaccine, a LSDV vaccine and two combined.
Overall, the study is interesting. The methods are described in sufficient detail to enable the study to be replicated.
The results are supported by the presented data, though I have made some suggestions below.
Similarly, the conclusions drawn are supported by the presented data.
My main comment on this manuscript would be the use of acute phase proteins (APP) as an indicator of the “acute immune response” as suggested by the manuscript title. Typically, AAP are used as indicators of inflammation and/or disease, rather than measures of the immune response.
Are the authors equating the “acute immune response” with “innate immune response”? If so I wonder why they have elected not to use the term “innate”.
Typically, when evaluating the innate immune response, we would measure pro-inflammatory cytokines and interferons. Indeed, one of the key advantages of a modified live vaccine is the potential to elicit a rapid non-specific anti-viral response. Did the authors consider measuring these? Eg Interferon-g, IL-6, TNF-a, IL-1 etc.
Long term inflammation is generally considered to be undesirable with respect to adaptive immunity. This might be the major consider with administering the two vaccines simultaneously.
Following the description of the three treatment groups, the authors subsequently refer to the groups as Group 1, Group 2 and Group 3. I would suggest the authors consider revising these group names to something more meaningful with respect to the treatments they receive. For those close to the study Group 1, 2, and 3 are meaningful, for those of us less familiar with the study, it means continuous checking back to make sure of what the treatment was. Perhaps something along the lines of Group 1 (FMDV), Group 2 (LSDV), and Group 3 (FMDV/LSDV). Or G1-FMDV, G2-LSD), and G3-FMDV/LSDV).
Line 61, 78 suggest using the preferred name of Türkiye for this country throughout the manuscript.
Line 72 How were the different breeds distributed between the treatment groups?
Consider adding a table which describes the various treatments, number of animals for each breed.
Did the authors record any other clinical parameters? Eg rectal temperature, behaviour changes etc. Where injection site inspected?
Line 114 Figure 1
The resolution of this figure is poor and should be improved.
The “title” of the graph should be the title of the Y-axis.
The treatments of the respective groups should be described in the figure legend.
Line 119 Figure 2 – same comments as per Figure 1.
I would also suggest that the data tables for Figure 1 and Figure 2 be supplied as a supplemental file.
Line 151 suggest replacing “symptoms” with “clinical signs” here and elsewhere as appropriate.
Line 157 The statement is correct in that the LSD vaccine is an attenuated virus vaccine. However, I do not believe the lactalbumin hydrolysate and sucrose are adjuvants. I think these are cryoprotectants/stabilisers used to ensure the virus is not inactivated in the freeze-drying processes. I could not find any mention of either in the cited reference 12. In most cases modified live viral vaccines do not contain adjuvants.
Please review and revise as appropriate.
Author Response
We really appreciate your detailed and kind comments, and it was very valuable for making our manuscript more refined.
Answers to your comments are provided with explanations below. In addition, we thought we need to revise the matters you mentioned. After revising what you said, it seems to be a better script.
We would like to express our thanks to you for taking the time to consider and advise on our manuscript.
[ *The approximate location of the revision is given below each comment and corrections in the text are underlined.]

Reviewer 2 Report
Comments and Suggestions for Authors
The manuscript titled "Analysis of Acute Immune Response Using Acute Phase Proteins Following Simultaneous Vaccination of Lumpy Skin Disease and Foot and Mouth Disease" describes the presence of acute phase proteins after vaccination with LSD and/or FMD vaccine in cattle. The findings are interesting and important for understanding the immediate effects vaccination may have on cattle and potential economical/agricultural impacts. Overall, the manuscript is well written and organized. The manuscript is accessible to a general audience. This study demonstrates the importance of carefully selecting vaccine types and adjuvant types to minimize adverse effects after vaccination. I have only minor comments to share:
Introduction: It would benefit to add a statement on acute phase proteins. What do they do? Why are they important to monitor post vaccination? What APPs will be monitored in the study?
Results
Line 103: Add space between GROUP1.
Figures 1 and 2 are pixelated and fuzzy. Please improve on the image quality of the two graphs.
Line 119: This should be changed from Figure 1 to Figure 2.
Author Response
We are grateful for your comments to make our manuscript more valuable. Thanks to your comments, through the various points you pointed out, we explained the experimental method and results in more clear and detailed description. Please review other changes as well. Thank you again for taking the time to review my manuscript.
[ *The approximate location of the revision is given below each comment and corrections in the text are marked in red.]
